# Inpainting Saturation Artifact in Anterior Segment Optical Coherence Tomography

**DOI:** 10.3390/s23239439

**Published:** 2023-11-27

**Authors:** Jie Li, He Zhang, Xiaoli Wang, Haoming Wang, Jingzi Hao, Guanhua Bai

**Affiliations:** Electronics Information Engineering College, Changchun Univesity, Changchun 130022, China; lij69@ccu.edu.cn (J.L.); wangxl@ccu.edu.cn (X.W.); 210401124@mails.ccu.edu.cn (H.W.); 210401143@mails.ccu.edu.cn (J.H.); 210401111@mails.ccu.edu.cn (G.B.)

**Keywords:** AS-OCT, corneal segmentation, inpainting, saturation artifact

## Abstract

The cornea is an important refractive structure in the human eye. The corneal segmentation technique provides valuable information for clinical diagnoses, such as corneal thickness. Non-contact anterior segment optical coherence tomography (AS-OCT) is a prevalent ophthalmic imaging technique that can visualize the anterior and posterior surfaces of the cornea. Nonetheless, during the imaging process, saturation artifacts are commonly generated due to the tangent of the corneal surface at that point, which is normal to the incident light source. This stripe-shaped saturation artifact covers the corneal surface, causing blurring of the corneal edge, reducing the accuracy of corneal segmentation. To settle this matter, an inpainting method that introduces structural similarity and frequency loss is proposed to remove the saturation artifact in AS-OCT images. Specifically, the structural similarity loss reconstructs the corneal structure and restores corneal textural details. The frequency loss combines the spatial domain with the frequency domain to ensure the overall consistency of the image in both domains. Furthermore, the performance of the proposed method in corneal segmentation tasks is evaluated, and the results indicate a significant benefit for subsequent clinical analysis.

## 1. Introduction

The cornea is an important refractive medium in the human eye, participating in the transmission and processing of visual information. The corneal segmentation technique provides support for the diagnosis and treatment of corneal diseases. Accurate segmentation is crucial, as a few micrometers of corneal segmentation errors can lead to significant changes in the derived clinical parameters [1]. Through corneal segmentation, the precise corneal morphology information [2] can be provided for the diagnosis and evaluation of diseases such as keratoconus [3], while also supporting preoperative preparation for procedures such as refractive surgery [1]. Anterior segment optical coherence tomography (AS-OCT) is a non-invasive imaging technique with a range of potential clinical applications [4,5], which provides high-resolution images of the anterior segment at the micron-scale resolution. AS-OCT achieves the resolution of the anterior and posterior surfaces of the entire cornea, as shown in Figure 1. However, during the imaging process, a strip-like saturation artifact (Figure 1a) is commonly generated, which covers the corneal surface and blurs the corneal edge. This is because the incident light source’s behavior at that point is normal to the tangent of the corneal surface. This high-intensity and high-contrast artifact negatively impacts the accuracy of corneal segmentation, making it challenging to obtain precise corneal morphology information. To improve the reliability and accuracy of corneal segmentation in various clinical applications and research studies, it is crucial to develop an inpainting method to eliminate the effects of the saturation artifact [6].

Recently, deep learning technology [7] has shown superiority in fields such as image processing [8]. AS-OCT image inpainting refers to the process of reconstructing the saturation artifact region and simultaneously maintaining the overall consistency of the image. Deep generative methods [9,10,11,12,13,14] are currently dominant, which effectively extract meaningful semantics from the image to be repaired and recover reasonable content with high visual fidelity, due to their powerful feature learning ability. Although these methods have achieved good performance with natural images, there are some differences between AS-OCT images and natural data, including the structure, noise, resolution, etc. These differences make it difficult to apply natural inpainting methods directly to AS-OCT saturation artifact restoration. AS-OCT inpainting needs to recover lost details more precisely to achieve the texture rationality and biological characteristics of eye tissue, such as the posterior surface of Bowman’s layer (BL).

Nowadays, many methods have been proposed for OCT image restoration. In order to avoid the problem of inaccurate quantification of the choroidal vascular system caused by retinal vascular shadows, Zhang et al. [15] proposed a three-stage image-processing framework to remove shadows in the retina, which may not be suitable for diseases with abnormal retinal pigment epithelial cell (RPE) layers. Cheong et al. [16] recommended the DeshadowGAN method for optical coherence tomography images of the optic nerve head vascular shadow, which performed well on healthy-eye OCT images. However, it is uncertain whether it can achieve the same performance in eyes with pathological conditions such as glaucoma. Liu et al. [17] recently projected a dictionary-based sparse representation method for saturation artifact removal in OCT images. Although this method works very well on narrow-range saturation artifacts, it can be less effective when presented in large areas of shadow. To solve this problem, Tang et al. [18] proposed a multi-scale framework for shadow repair to achieve the repair effect of both wide and narrow retinal vascular shadows, but this repair effect on real shadows cannot be consistent with the composite shadow restoration effect. Nevertheless, these methods are proposed for unique OCT images collected from the various OCT devices. Due to the different data characteristics collected using different OCT devices, it is challenging to directly use existing OCT repair methods for saturation artifact restoration in AS-OCT images. There is an urgent for a dedicated inpainting method to remove saturation artifact in AS-OCT images.

To solve the issues of artifacts and speckle noise patterns and precisely segment the shallowest tissue interface in an AS-OCT image, Ouyang et al. [6] proposed a cascaded neural network framework. However, this frame only removes saturation artifacts and speckle noise patterns just above the shallowest tissue interface via cGAN [19], and does not completely remove the saturation artifact penetrating the center of the cornea. Recently, in order to address the issue of stripe artifacts damaging the visual quality of images and affecting automatic ophthalmic analysis, Bai et al. [20] proposed SC_GAN, which can successfully removed artifacts but cannot significantly improve clinical segmentation accuracy.

In this paper, we treat saturation artifact inpainting as converting the artifact image into an artifact-free image. A new generative model is proposed to inpaint the corneal structure and texture obscured by saturation artifacts, as well as maintaining the overall consistency of the image. The main novelties and contributions are as follows:The dual-domain transformation capability of DualGAN [21] is designed to achieve AS-OCT saturation artifact inpainting by converting the artifact image into an artifact-free image. The structural similarity loss for reconstructing the structure and texture of the cornea is incorporated;A frequency loss that combines the spatial and frequency domains is introduced to ensure the overall consistency of the images in both domains;The repair experiments on both synthetic and real artifacts are devised. The results indicate that the proposed methods can restore artifacts in different situations. To confirm the clinical value of saturation artifact inpainting, segmentation experiments are designed on the three corneal boundaries of real artifact-inpainted images, including the anterior surface of the epithelium (EP), the posterior surface of Bowman’s layer (BL), and the posterior surface of the endothelium (EN). The experimental results demonstrate that the method significantly enhances the precision of corneal segmentation, proving to be more accurate than other repair techniques.

The remainder of the paper is organized as follows: We introduce our approach in Section 2. In Section 3, the experiments are described, specifically for synthesis and real artifact restoration (Section 3.3), segmentation experiments for real restoration verification (Section 3.4), and ablation experiments (Section 3.5), followed by the conclusions in Section 4. 

## 2. Proposed Method

In this work, AS-OCT image inpainting is regarded as a translation task from artifact images to artifact-free data. As shown in Figure 2, the proposed method is built on DualGAN, including two generators ($G_{A}$, $G_{B}$) and two discriminators ($D_{A}$, $D_{B}$), and introduces two loss terms: structural similarity and frequency loss. By giving two paired images sampled from domains *S* and *W*, the forward task of the network is to learn the generator $G_{A}$: *S → W* that maps the images s∈S to w∈W, and the backward task is to train the generator $G_{B}$: *W → S*. The generated image and ground_truth image are constrained using four different loss functions, including $L_{F}$, $L_{SSIM}$, $L_{adv}$, and $L_{1}$, making the former more similar to the latter. $L_{F}$ combines the spatial and frequency domain of the image. Moreover, the reconstruction loss, $L_{recon},$ further restrains the reconstructed image to be consistent with the ground_truth. Two discriminators are used to evaluate the fit between the output of the corresponding generator and the ground_truth. In this section, the generator, discriminator, and loss functions are described in detail.

### 2.1. Network Architecture

The network consists of two generators ($G_{A}$, $G_{B}$) and two discriminators ($D_{A}$, $D_{B}$). As shown in Figure 3, the two generators achieve mutual conversion between *S-domain* and *W-domain* images, and the two discriminators are used to distinguish the generated image and real data. Generator $G_{A}$ generates artifact-free images, while $G_{B}$ generates an artifact image with a central stripe pattern. Discriminator $D_{A}$ is trained to distinguish between real samples in the *W-domain* and data generated from the *S-domain*, while $D_{B}$ is trained in the opposite manner.

Both generators, $G_{A}$ and $G_{B},$ employ the U-Net [22] architecture with eight convolutional layers for the encoder and decoder, as shown in Figure 4a. The encoder consists of a convolutional layer, six Relu-Conv-LayerNorm (RCL) blocks, and one Relu-Conv (RC) block. The decoder comprises seven Relu-ConvTranspose-BatchNorm-Dropout (RCBD) blocks and one Relu-ConvTranspose-Tanh (RCT) block. The encoder and decoder are connected through skip connections.

Both discriminators, $D_{A}$ and $D_{B},$ use the PatchGAN [23] structure, as shown in Figure 4b. The receptive field of PatchGAN is 70 × 70, and it consists of five 4 × 4 convolutional layers. Specifically, it includes a Conv-LeakyRelu (CL) block, three Conv-BatchNorm-LeakyRelu (CBL) blocks, and a Conv-Sigmoid (CS) block. 

### 2.2. Objective Functions

Formally, image s∈S is input into generator $G_{A}$ to derive the image $G_{A}\left(s\right)\in W$, and the image is processed through $G_{B}$ to obtain the reconstructed image, $G_{B}(G_{A}\left(s\right))\in S$. Similarly, image w∈W is passed through generator $G_{B}$ to achieve the image $G_{B}(w)\in S$, which is input into $G_{A}$ to obtain the reconstructed image, $G_{A}(G_{B}\left(w\right))\in W$. Discriminator $D_{A}$ evaluates the degree of fit between $G_{A}\left(s\right)$ and the real w. Discriminator $D_{B}$ estimates the measure of fit between the fake s generated with $G_{B}$ and the real s. 

In practice, generator $G_{A}$ generates the image $\hat{s}$. In order to avoid network damage to other organizational structures of image s, the images, s and $\hat{s}$, and the corresponding mask (m) were integrated; thus, the inpainted results can be denoted as follows:(1)$$G_{A}(s)=s\;ʘ(1-m)+\hat{s}ʘm$$
where ʘ is the pixel-wise multiplication. Equation (1) ensures that image s is only replaced by the generated image in the artifact region, while other regions remain unchanged.

**SSIM Loss:** Although DualGAN achieves image translation between two domains, it distorts the corneal boundary structure and texture information in AS-OCT images. Structural similarity (SSIM) [24] is a powerful tool for image quality assessment. Generally, a higher SSIM means that the image has clearer results. SSIM has been widely used in tasks such as image restoration [25], semantic segmentation [26], and dehazing [27] since it was proposed. Therefore, SSIM is adopted as a loss function to train the network to reconstruct the corneal structure and can improve the corneal segmentation accuracy. Since SSIM is pixel-based, this loss function has constraints on corneal structure reconstruction and helps to restore texture information. *SSIM* for two images, *x* and *y* (*x* is the ground_truth, *y* denotes the repaired image), is defined as follows:(2)$$SSIM(x,y)=\frac{\left(2\mu_{x}\mu_{y}+C_{1})(2\sigma_{xy}+C_{2}\right)}{\left(\mu_{x}^{2}+\mu_{y}^{2}+C_{1})(\sigma_{x}^{2}+\sigma_{y}^{2}+C_{2}\right)}$$
where *µ* and *σ* represent the standard deviation and covariance of images, respectively. $C_{1}$ and $C_{2}$ are constants. A higher SSIM indicates that the two images are more similar to each other, and the SSIM equals 1 for identical images. The loss function for the SSIM operates on both generators and can then be written as follows:(3)$$L_{SSIM}=E_{W}\left[1-SSIM\left(w,G_{A}\left(s\right)\right)\right]+E_{S}\left[1-SSIM\left(s,G_{B}\left(w\right)\right)\right]$$

**Frequency Loss:** In AS-OCT images, non-tissue structure saturation artifact is usually positioned at high frequencies, which could overshadow or interfere with other frequency components. There are gaps between the real and generated images, especially in the frequency domain [28,29], as shown in Figure 5. For different images, their frequency domain distribution also varies. Figure 5b,c, respectively, show the frequency domain distribution of synthesized and real saturated artifact images, which have significant differences from the that of artifact-free images. Compared to Figure 5a, these have significant changes in the horizontal direction in the spatial domain, so their frequency domain distribution has a bright line in the horizontal direction, especially in Figure 5b, and the brightness in the central low-frequency region is also different.

To minimize the disparities, the frequency loss is introduced, which is the main idea of obtaining frequency domain characteristics of the real and generated images by performing a fast Fourier transform (FFT); in other words, make Figure 5d infinitely close to Figure 5a to achieve the same. Previous studies [30,31] found it beneficial to replace distance $L_{2}$ with distance $L_{1},$ since the former often leads to blurriness. The distance formulas for $L_{1}$ and $L_{2}$ are Equations (4) and (5), respectively. Hence, the $L_{1}$ distance is adopted to measure the distance between the real image and the generated image after FFT. The spatial and frequency domains are combined to further improve the quality of restoration. The frequency loss function operates on both generators and can then be written as Equation (6).
(4)$${\left||L|\right|}_{1}=\sum_{i=1}^{n}\left|x_{i}-y_{i}\right|$$
(5)$${\left||L|\right|}_{2}=\sum_{i=1}^{n}\sqrt{{\left(x_{i}-y_{i}\right)}^{2}}$$
(6)$$L_{F}=E_{W}\left[{\left|\left|F(G_{A}(s))-F(w)\right|\right|}_{1}\right]+E_{S}[{\left||F(G_{B}(w))-F(s)|\right|}_{1}]$$
where *F* represents the fast Fourier transform, transforming the image from the spatial distribution to the frequency domain via FFT.

$L_{1}$ **Loss:** we use $L_{1}$ loss to measure the difference between the generated image and the real image, to avoid some pixels being smoothed by transitions, resulting in the resulting image missing detail and texture information. $L_{1}$ loss is defined as
(7)$$L_{1}=E_{W}\left[{\left|\left|G_{A}(s)-w\right|\right|}_{1}\right]+E_{S}[{\left||G_{B}(w)-s|\right|}_{1}]$$

**Adversarial Loss:** the adversarial loss acts between two pairs of generators–discriminators ($G_{A}-D_{A},\;G_{B}-D_{B}$), and is defined as
(8)$$\begin{array}{c} L_{adv}\left(G_{A},D_{A},G_{B},D_{B}\right)=E_{w\epsilon W}\left[logD_{A}\left(w\right)\right]+E_{s\epsilon S}[log(1-D_{A}(G_{A}(s)))]\\ +E_{s\epsilon S}[logD_{B}(s)]+E_{w\epsilon W}[log(1-D_{B}(G_{B}(w)))] \end{array}$$

Discriminator $D_{A}$ is trained with the real w as positive samples and the generated $G_{A}(s)$ as negative examples, whereas $D_{B}$ takes the real s as positive and $G_{B}(w)$ as negative. Generators $G_{A}$ and $G_{B}$ are optimized to emulate ‘fake’ outputs to blind the corresponding discriminators, $D_{A}$ and $D_{B}$.

**Reconstruction Loss:** the $L_{1}$ distance between the reconstructed image and the real image is adopted as the reconstruction loss, formulated as
(9)$$L_{recom}=E_{s\epsilon S}\left[{\left||G_{B}\left(G_{A}\left(s\right)\right)-s|\right|}_{1}\right]+E_{w\epsilon W}[{\left||G_{A}\left(G_{B}\left(w\right)\right)-w|\right|}_{1}]$$

**Total Losses:** the whole objective function of the proposed network can be written as
(10)$$L_{total}=\lambda_{SSIM}L_{SSIM}+\lambda_{F}L_{F}+\lambda_{1}L_{1}+\lambda_{adv}L_{adv}+\lambda_{recon}L_{recom}$$
where $\lambda_{SSIM}$, $\lambda_{F}$, $\lambda_{1}$, $\lambda_{adv},$ and $\lambda_{recon}$ are the tradeoff parameters, and set 50, 1, 100, 1, and 1, respectively.

## 3. Experimental Setup and Results

In this section, the details of data preprocessing are presented in Section 3.1, and some parameter settings during the training process are explained in Section 3.2. Quantitative and qualitative evaluations of the synthetic artifact restoration are conducted in Section 3.3, and a visualization of real restoration is performed. Then, in Section 3.4, the real repair quality is verified through corneal segmentation experiments. Section 3.5 introduces extensive ablation studies to validate the effectiveness of various components of the model.

### 3.1. Data Preprocessing

The AS-OCT image used in this article is from a CASIA1 [32] ophthalmology device (Tomey Inc., Nagoya, Japan), using a swept-source OCT (SS-OCT) with a scanning speed of 30,000 A-ultrasound/second, a wavelength of 1310 nm, and a frequency of 50 Hz, whilst preserving an original image size of 1689×1000. To more accurately repair saturation artifacts throughout the corneal tissue, the AS-OCT image is cropped to the size of 256×256 to obtain the image, *I*. 

Since saturation artifact images do not have paired data, the binary mask (m) (with value 0 for known pixels and 1 for the area to be repaired) is manually added to artifact-free image *I* with varying degrees of inclination in the corneal tissue to simulate saturation artifacts in the spatial domain. Figure 6 shows the process of obtaining the synthetic artifact image s=I×m, where the position and width of the mask are different. The paired data for image s are the artifact-free image w=I. A total of 3774 pairs of synthetic images were obtained; hence, 3374 were employed as a training dataset, while the remaining subsets served as test set. To verify the effectiveness of the network, the mask in the test set was wider than the mask added in the training set, and 80 AS-OCT real artifact images were tested.

### 3.2. Training Parameters

All training and testing is performed on a single NVIDIA GeForce RTX 3090 GPU (24 GB). Using the RMSprop optimizer, the learning rates of the generator and discriminator are 0.0005, and 0.0001, respectively, and the batch size is 4.

### 3.3. Evaluation of Inpainting

To validate the proposed method, the model is applied for a comparison with existing deep-learning inpainting methods, including MADF [9], CTSDG [10], AOT-GAN [11], RFR [12], G&L [13], PICNet [14], and DualGAN [21].

**Synthetic Saturation Artifacts’ Qualitative Comparison:** the comparative experiments on synthetic artifact restoration are conducted in three different situations:The repair effects of different methods on corneal tissue images with different tilt degrees under the same mask conditions are shown in Figure 7;The inpainting results of different methods on corneal tissue images with the same inclination combined with different masks are shown in Figure 8. Briefly, Figure 8 shows the image restoration effects of three groups, adding different masks to the same AS-OCT image. Figure 8(I), (II), and (III) respectively show the repair results of images with downward tilt of corneal tissue, images with no tilt degree of corneal tissue, and images with upward tilt of corneal tissue with different masks added.The results of adding different masks into the corneal tissue images with different degrees of inclination using different methods are shown in Figure 9.

As shown in these renderings, the MADF method can achieve non-tilted corneal tissue image restoration, but there may be some structural defects for wide shadows and tilted corneal tissue images. For CTSDG, AOT-GAN, and G&L, these methods can repair the corneal structure, but they retain blurry or overly smooth textures in shaded areas, especially in wide-shaded images. The RFR approach may effectively rectify backdrop pixels, yet manifests a limited capacity when addressing the corneal structure and texture. The PICNet method has a certain repair effect on the mask area, but its processing ability of the narrow-band artifact’s edge is significantly weak, resulting in incomplete restoration. Significate repair marks occur with the DualGAN method, which can distort the corneal structure and texture. The proposed method can successfully rectify artifacts across varying widths and positions, irrespective of the degree of inclination of the corneal tissue.

**Synthetic Saturation Artifacts’ Quantitative Comparison:** To quantitatively evaluate the inpainting quality, the means of the peak signal-to-noise ratio (PSNR), SSIM, and learned perceptual image patch similarity (LPIPS) [33] are calculated for the repaired regions from different techniques on the synthetic testing set. Table 1 shows the results, with the best performance value written in bold. For CTSDG, AOT-GAN, and G&L, these methods add noise beyond the original image to the repair results. As the PSNR is more sensitive to noise, these methods achieve higher values in this metric. In addition, the SSIM values indicate that these methods have a positive effect on the reconstruction of corneal structures. The RFR method is unable to repair the structure and texture of the cornea, resulting in low values for various indicators. The MADF, PICNet, and DualGAN methods have varying degrees of inpainting effects, but due to the distortion of the corneal structure in the repair results, low SSIM values are achieved. Although our method cannot achieve optimal results in the PSNR and SSIM metrics for synthetic saturated artifacts, the image quality is also crucial for corneal boundary segmentation [34]. Therefore, it is necessary to pay attention to the corneal tissue structure and texture information simultaneously. LPIPS can better simulate human visual perception and subjective quality judgment. For synthetic artifact restoration, the recommended method achieves the best performance in LPIPS.

**Real Saturation Artifact Qualitative Comparison:** Figure 10 shows the inpainting performance of different methods for real artifacts, and these comparative methods show consistent repair results with the synthetic saturation artifacts. Specifically, the MADF method can repair narrow-range artifacts, while repairing wide-range artifacts can lead to corneal structural protrusions. The CTSDG, AOT-GAN, and G&L methods bring blurry or smooth texture details, regardless of the wideness of the artifacts. The RFR method can only play a certain role in repairing background pixels. The PICNet method performs well in wide artifact restoration, but it may have blurring for narrow-range artifacts and has weak edge-restoration capabilities. The baseline network, DualGAN, distorts the corneal structure. Our model can repair saturation artifacts with different widths and positions, both synthetic and real, reconstructing the corneal structures and repairing textural details.

### 3.4. Evaluation on Segmentation

To further evaluate the inpainting performance of different methods on machine analysis, experiments on a corneal segmentation task are conducted. Concretely, the widely used U-Net segmentation model was trained using the labelled AS-OCT image; 31 AS-OCT images are randomly selected from the real repaired saturation artifact to test the segmentation effect. The inpainting and segmentation results of each method are shown in Figure 11. The visual segmentation effect shows that the proposed method maintains overall image consistency while removing the saturation artifact and reconstructing the corneal structure and textural details, due to the constraints of structural similarity and frequency domain loss. In addition, the performance of different inpainting methods is analyzed by calculating the Dice similarity coefficient (*DSC*), pixel accuracy (*PA*), F1-score, and Jaccard value of the segmentation results. The calculation formulas for DSC, PA, F1-score, and Jaccard are shown in Equations (11)–(14).
(11)$$DSC=\frac{2|X\cap Y|}{\left|X|+|Y\right|}\;$$
where *X* and *Y* represent the generated image and the ground_truth image, respectively, and X∩Y is the intersection of the two.
(12)$$PA=\frac{TP+TN}{TP+TN+FP+FN}$$
where *TP* is the number of true positives, *TN* is the number of true negatives, *FN* is the number of false negatives, and *FP* is the number of false positives.
(13)$$F_{1}=\frac{2TP}{2TP+FN+FP}$$
(14)$$Jaccard=\frac{\left|X\cap Y\right|}{\left|X\cup Y\right|}$$
where X∪Y signals the union of the generated image and the ground_true.

As shown in Table 2, our method is ahead of other methods in calculating various indicators, especially DSC, with a maximum DSC value of 0.732, which is not presented in other methods. The average values calculated for each indicator are provided in the table, indicating that our inpainting method has significantly improved the corneal segmentation.

### 3.5. Ablation Studies

In this section, two sets of experiments are conducted to verify the contribution. Here, the impact of structural similarity loss and frequency loss on the repair of saturation artifacts are studied. Specifically, the structural similarity and frequency losses are removed, respectively, and their impact on the experiment is observed. From the results in Figure 12, it can be seen that removing the structural loss will seriously affect the texture and structural information, especially in Bowman’s layered structure. Meanwhile, the segmentation indicators in Table 3 also show low segmentation accuracy due to the incomplete corneal structure. Unlike removing structural loss, although removing frequency loss can maintain almost the same visual effect as the suggested method, Table 3 also indicates that adding frequency loss can improve the accuracy of corneal segmentation. In summary, the results reveal the role of structural similarity loss and frequency loss in the repair of saturation artifacts, and also confirm the superiority of the proposed method.

## 4. Conclusions

In this study, a novel deep generative method is proposed to remove saturation artifacts generated during AS-OCT imaging. The proposed model is a bilateral network and introduces two loss terms, namely structural similarity loss and frequency loss. The structural similarity loss reconstructs structural and textural details while ensuring the overall image consistency. In the frequency domain, the saturated artifact is located in the high-frequency component, and frequency loss is designed to eliminate image distortion in the high-frequency region. Moreover, frequency loss combines the spatial domain with the frequency domain to reduce the gap between the generated image and the real image. The repair comparison experiment and ablation experiment demonstrate the effectiveness of the innovative points. The results show that this method can successfully remove saturation artifacts and restore the structure and texture information of the cornea. The corneal boundary segmentation experiment further verifies that the proposed method effectively improves the quality of AS-OCT images and significantly improves the accuracy of corneal segmentation.

Although this method has achieved satisfactory results, there are still some limitations to the framework. Specific device data are adopted to obtain these valuable results, while the image restoration performance of other OCT devices is not determined in this study.

In summary, an inpainting method was proposed to remove AS-OCT saturation artifact while preserving the surroundings. The proposed network is of great significance for the improvement of corneal segmentation accuracy.

## Figures and Tables

**Figure 1 sensors-23-09439-f001:**
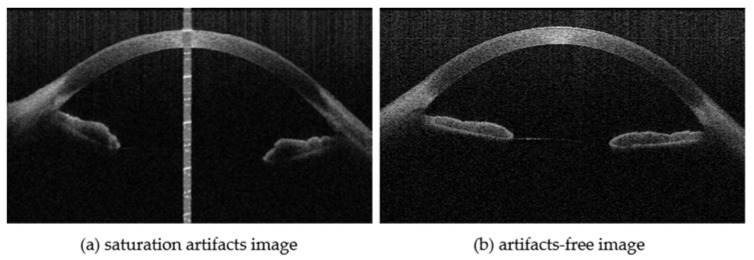
AS-OCT can visualize the entire cornea.

**Figure 2 sensors-23-09439-f002:**
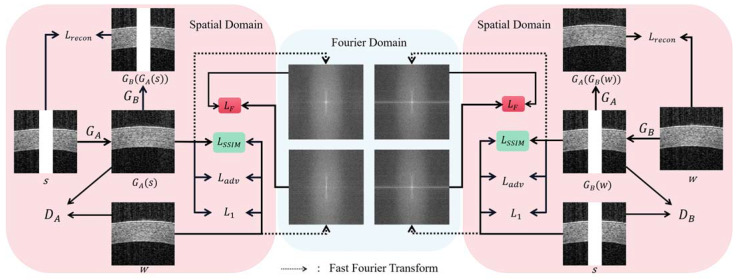
The overall architecture of the proposed model includes two generators ($G_{A}$, $G_{B}$) and two discriminators ($D_{A}$, $D_{B}$). Image s∈S is input to generator $G_{A}$ and image w∈W is input to generator $G_{B}$. The proposed structural similarity loss and frequency loss are represented as $L_{SSIM}$ and $L_{F}$, respectively. $L_{F}$ combines spatial and frequency domains.

**Figure 3 sensors-23-09439-f003:**
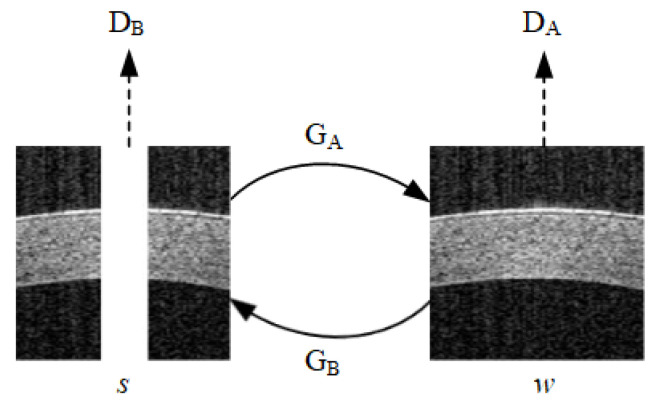
The mutual conversion between *S-domain* and *W-domain* images through two generators. Two discriminators are used to discern between generated images and real data.

**Figure 4 sensors-23-09439-f004:**
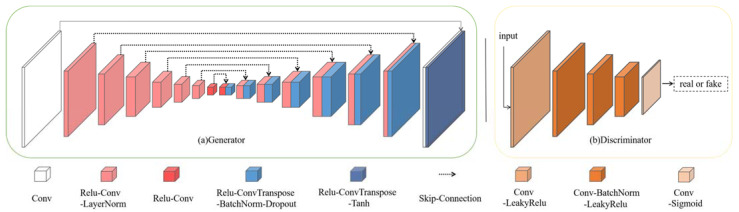
The proposed method structure of generators and discriminators. The generators adopt a U-Net architecture, and the discriminators employ PatchGAN with five convolutional layers.

**Figure 5 sensors-23-09439-f005:**
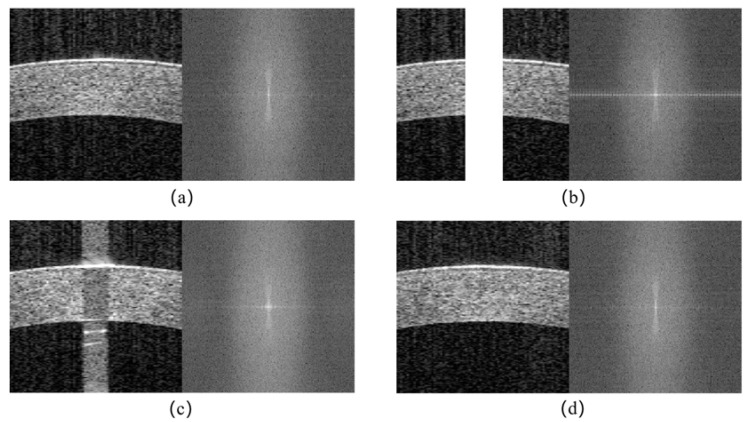
Feature distribution of different images in space and frequency domains: (**a**) artifact-free image; (**b**) synthetic saturation artifact image; (**c**) real saturation artifact image; (**d**) after inpainting data.

**Figure 6 sensors-23-09439-f006:**
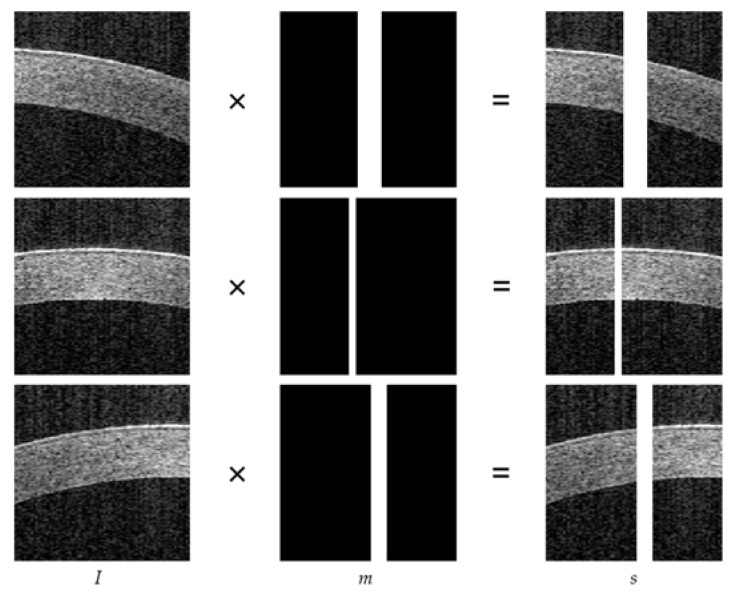
The process of creating the synthesized saturated artifact image, s=I×m.

**Figure 7 sensors-23-09439-f007:**
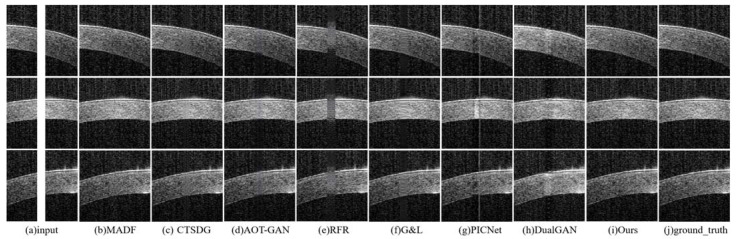
Experimental results of synthetic saturation artifact repair using corneal tissues with different slopes combined with the same mask: (**a**) synthetic saturation artifact input; (**b**) MADF; (**c**) CTSDG; (**d**) AOT-GAN; (**e**) RFR; (**f**) G&L; (**g**) PICNet; (**h**) DualGAN; (**i**) ours; and (**j**) ground_truth.

**Figure 8 sensors-23-09439-f008:**
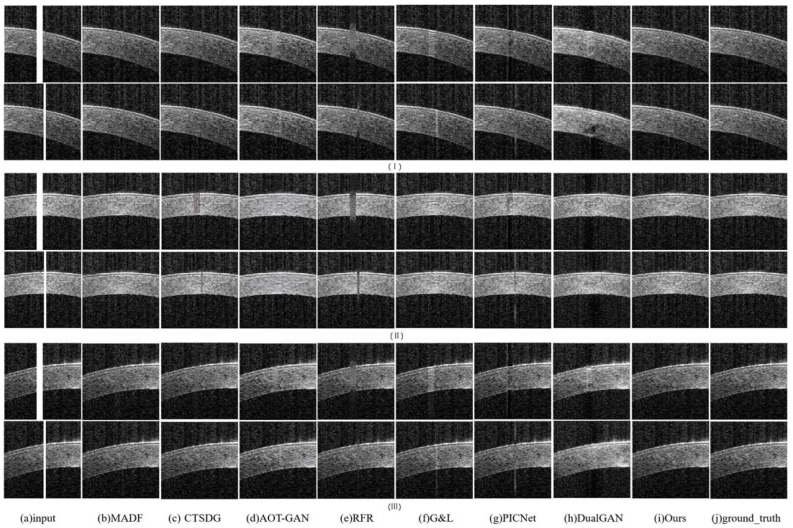
Three groups of experimental results of synthetic saturation artifact inpainting, using corneal tissue with the same inclination degree combined with different masks: (**a**) synthetic saturation artifact input; (**b**) MADF; (**c**) CTSDG; (**d**) AOT-GAN; (**e**) RFR; (**f**) G&L; (**g**) PICNet; (**h**) DualGAN; (**i**) ours; and (**j**) ground_truth.

**Figure 9 sensors-23-09439-f009:**
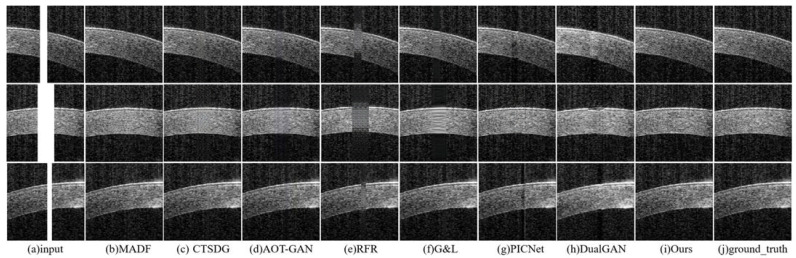
Experimental results of synthetic saturation artifact restoration, using corneal tissue with different tilt angles combined with different masks: (**a**) synthetic saturation artifact input; (**b**) MADF; (**c**) CTSDG; (**d**) AOT-GAN; (**e**) RFR; (**f**) G&L; (**g**) PICNet; (**h**) DualGAN; (**i**) ours; and (**j**) ground_truth.

**Figure 10 sensors-23-09439-f010:**
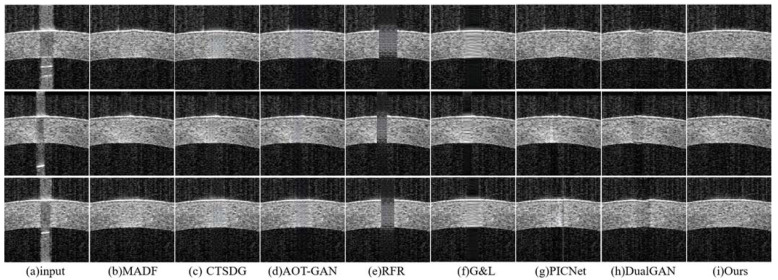
Experimental results for real saturation artifacts: (**a**) saturation artifact input; (**b**) MADF; (**c**) CTSDG; (**d**) AOT-GAN; (**e**) RFR; (**f**) G&L; (**g**) PICNet; (**h**) DualGAN; and (**i**) ours.

**Figure 11 sensors-23-09439-f011:**
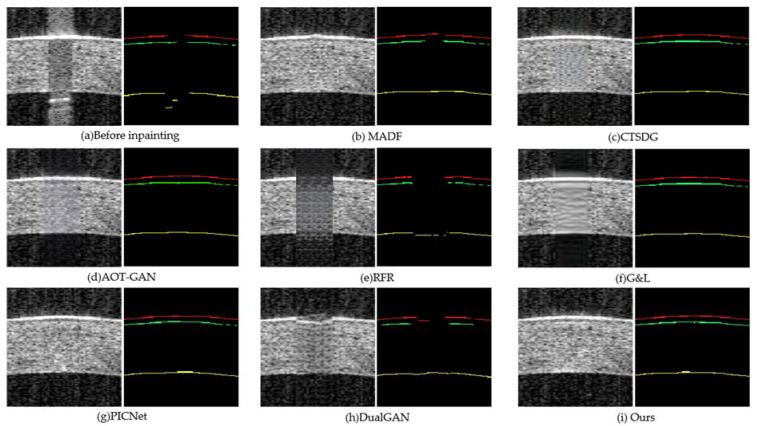
The inpainting and segmentation results: (**a**) saturation artifact image before inpainting; (**b**) MADF; (**c**) CTSDG; (**d**) AOT-GAN; (**e**) RFR; (**f**) G&L; (**g**) PICNet; (**h**) DualGAN; and (**i**) ours. EP, BL, and EN are represented by red, green, and yellow lines, respectively.

**Figure 12 sensors-23-09439-f012:**
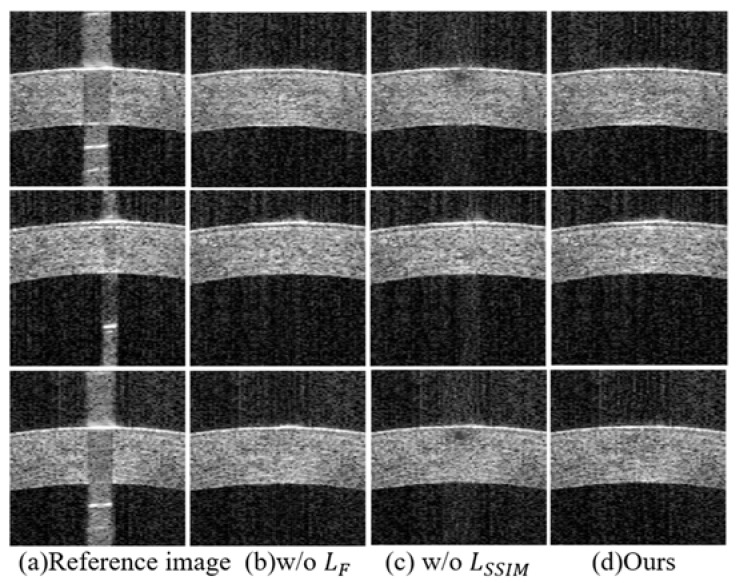
The ablation study results: (**a**) saturation artifact image; (**b**) w/o $L_{F}$; (**c**) w/o $L_{SSIM}$; (**d**) ours.

**Table 1 sensors-23-09439-t001:** Quantitative comparison of synthetic artifact inpainting results using different methods. ↑ Higher is better. ↓ Lower is better. The best results are presented in bold.

Method	PSNR↑	SSIM↑	LPIPS↓
Before Inpainting	4.899	0.314	0.687
MADF	20.533	0.392	0.091
CTSDG	22.886	0.554	0.080
AOT-GAN	**22.168**	0.518	0.125
RFR	16.168	0.431	0.213
G&L	22.879	**0.561**	0.225
PICNet	19.977	0.461	0.125
DualGAN	17.9154	0.250	0.106
Ours	21.333	0.496	**0.072**

**Table 2 sensors-23-09439-t002:** Quantitative comparison of corneal segmentation results in AS-OCT images using different methods. ↑ Higher is better. The best results are presented in bold.

Method	DSC↑	PA↑	F1-Score↑	Jaccard↑
Before Inpainting	0.431	0.986	0.984	0.299
MADF	0.494	0.987	0.985	0.357
CTSDG	0.536	0.988	0.986	0.390
AOT-GAN	0.520	0.987	0.985	0.381
RFR	0.409	0.986	0.985	0.280
G&L	0.531	0.988	0.986	0.387
PICNet	0.482	0.987	0.985	0.340
DualGAN	0.470	0.987	0.985	0.335
Ours	**0.585**	**0.989**	**0.987**	**0.444**

**Table 3 sensors-23-09439-t003:** Quantitative comparison of corneal segmentation results in ablation study. ↑ Higher is better. The best results are presented in bold.

Method	DSC↑	PA↑	F1-Score↑	Jaccard↑
w/o $L_{F}$	0.570	0.989	0.987	0.425
w/o $L_{SSIM}$	0.479	0.987	0.985	0.338
Ours	**0.585**	**0.989**	**0.987**	**0.444**

## Data Availability

The data are not publicly available due to the privacy of clinical data.

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
