# Peer review of "Inpainting Saturation Artifact in Anterior Segment Optical Coherence Tomography"

_sensors, 2023, doi:10.3390/s23239439_

Round 1
Reviewer 1 Report
Comments and Suggestions for Authors
The authors presents a segmentation technique for saturation artifacts with tomographic images of Cornea. The work offers significance in easing the work of ophthalmologist. However, the work requires some improvements before it could be published.
1- Some explanation regarding white mask is required. Is it applied in spatial domain or the frequency. What is its structure?
2- The authors use a generative network to regenerate the image, how do they ascertain that this technique will not compromise other anomalies in the image that can important for reaching a prognosis by an ophthalmologist.
3- What is the difference between Dice Similarity Coefficient and F1-Score.
4-Since it is a retrieval problem the authors can consider incorporating Precision-Recall metrics and graphs for the proposed and existing methods. This will help the reader to readily understand the performance of these methods.
5- The authors should explain in conclusion or discussion the theoretical basis on which the proposed model is able to perform better.
6- Since the work involves image processing and deep learning the author should consider citing
"Khan, Yaser Daanial; Mahmood, M Khalid; Ahmad, Daud; Al-Zidi, Nasser M; ",Content-Based Image Retrieval Using Gamma Distribution and Mixture Model,Journal of Function Spaces,2022,,,2022,Hindawi
Yang, Y., Gao, D., Xie, X., Qin, J., Li, J., Lin, H., ... & Deng, K. (2022). DeepIDC: a prediction framework of injectable drug combination based on heterogeneous information and deep learning. Clinical Pharmacokinetics, 61(12), 1749-1759.
Reviewer 2 Report
Comments and Suggestions for Authors
Please answer one question before my comments. What is the difference between the current work and "
A Structure-Consistency GAN for Unpaired AS-OCT Image Inpainting
"
Comments on the Quality of English LanguageModerate language issues were detected.
Reviewer 3 Report
Comments and Suggestions for Authors
In this manuscript the authors present an approach to correct specific saturation artifacts in AS-OCT images. Comments on the manuscript appear below
- The main issue of the manuscript has to do with the novelty of the results. Althought the specific approach might be somehow novel, the comparison with other similar approaches seem to provide slight differences in the quality of the results. What is more, the images are extracted from a commercial OCT setup, so there are probably several image treatment algorithms already applied to the images, that could influence the results.
- Due to the same reason, the applicability of the results could be limited, as there are already several approaches with similar results. This should be clearly described in the manuscript.
- The particular parameters employed in the commercial OCT system should be described in detail.
- The general structure of the manuscript is a bit confusing. The introduction lacks a description of the sections that follow. The discussion of the results is embedded in the results section, and some comparisons remain poorly explained.
- Distances L1 and L2 are not clearly defined in the explanation of frequency loss.
- What does the expression “…masks are wider…” mean in section 3.1?
- The authors seem to train the network only with synthetic artifacts images, why did you not to add real artifacts images in this phase?
- Figures 6, 7 and 8 are not discussed in detail.
- The results on real saturation artifacts are not analyzed quantitatively.
- A general language editing shoul be made, to correct mistakes such as:
o Abstract, “… the central artifacts is commonly…”
o Section 2, “…without saturated artifacts”
o …
- Figure 1 is not mentioned in the text
Figure 2 is not clearly and deeply explained.
Comments on the Quality of English LanguageA general review of English expressions must be made.
Round 2
Reviewer 2 Report
Comments and Suggestions for Authors
The authors addressed my concerns, the manuscript is good for publication.
Author Response
Dear Reviewer,
Thank you for your encouragement. It is a great honor to obtain your recognition of this work.
Reviewer 3 Report
Comments and Suggestions for Authors
The revised version of the manuscript addresses most of the comments exposed in the previous review round.
Comments on the Quality of English LanguageA slight review of English expressions must be made.
Author Response
Dear Reviewer,
Thank you very much for your suggestion. We have checked and revised the English expression of the paper.